# Deep Policy Dynamic Programming
# for Vehicle Routing Problems

## Abstract

Routing problems are a class of combinatorial problems with many practical
applications. Recently, end-to-end deep learning methods have been proposed
to learn approximate solution heuristics for such problems. In contrast, classical
dynamic programming (DP) algorithms guarantee optimal solutions, but scale
badly with the problem size. We propose *Deep Policy Dynamic Programming*
(DPDP), which aims to combine the strengths of learned neural heuristics with
those of DP algorithms. DPDP prioritizes and restricts the DP state space using
a policy derived from a deep neural network, which is trained to predict edges
from example solutions. We evaluate our framework on the travelling salesman
problem (TSP), the vehicle routing problem (VRP) and TSP with time windows
(TSPTW) and show that the neural policy improves the performance of (restricted)
DP algorithms, making them competitive to strong alternatives such as LKH, while
also outperforming most other 'neural approaches' for solving TSPs, VRPs and
TSPTWs with 100 nodes.

## 1 Introduction

Dynamic programming (DP) is a powerful framework for solving optimization problems by solving
smaller subproblems through the principle of optimality [3]. Famous examples are Dijkstra's
algorithm [14] for the shortest route between two locations, and the classic Held-Karp algorithm for
the travelling salesman problem (TSP) [23, 4]. Despite their long history, dynamic programming
algorithms for vehicle routing problems (VRPs) have seen limited use in practice, primarily due to
their bad scaling performance. More recently, a line of research has attempted the use of machine
learning (especially deep learning) to automatically learn heuristics for solving routing problems
[57, 5, 41, 29, 7]. While the results are promising, most learned heuristics are not (yet) competitive
to 'traditional' algorithms such as LKH [24] and lack (asymptotic) guarantees on their performance.

In this paper, we propose *Deep Policy Dynamic Programming* (DPDP) as a framework for solving
vehicle routing problems. The key of DPDP is to combine the strengths of deep learning and DP,
by restricting the DP state space (the search space) using a policy derived from a neural network.
In Figure 1 it can be seen how the neural network indicates promising parts of the search space
(through a *heatmap* over the edges of the graph), which is then used by the DP algorithm to find a
good solution. DPDP is more powerful than some related ideas [62, 52, 61, 6, 34] as it combines
supervised training of a large neural network with just a *single* model evaluation at test time, to enable
running a large scale guided search using DP. The DP framework is flexible as it can model a variety
of realistic routing problems with difficult practical constraints [20]. We illustrate this by testing
DPDP on the TSP, the capacitated VRP and the TSP with (hard) time window constraints (TSPTW).

In more detail, the starting point of our proposed approach is a *restricted dynamic programming*
algorithm [20]. Such an algorithm heuristically reduces the search space by retaining only the $B$ most

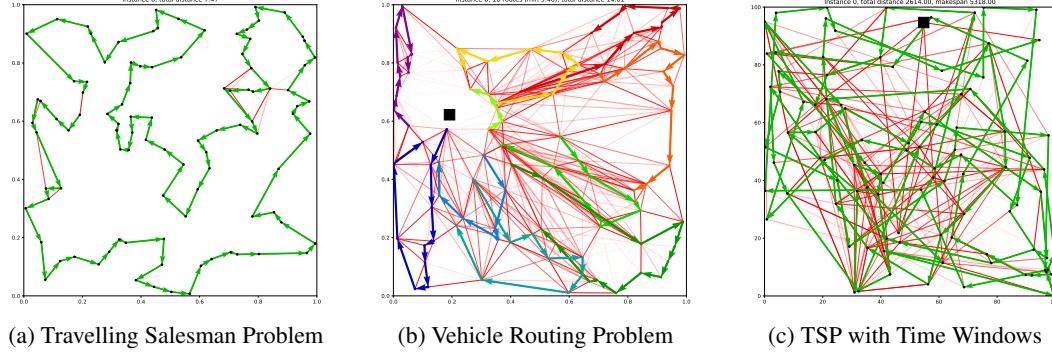

(a) Travelling Salesman Problem  (b) Vehicle Routing Problem  (c) TSP with Time Windows

Figure 1: Heatmap predictions (red) and solutions (colored) by DPDP (VRP depot edges omitted).

promising solutions per iteration. The selection process is very important as it defines the part of the DP state space considered and, thus, the quality of the solution found (see Fig. 2). Instead of manually defining a selection criterion, DPDP defines it using a (sparse) heatmap of promising route segments obtained by pre-processing the problem instance using a (deep) graph neural network (GNN) [26]. This approach is reminiscent of neural branching policies for branch-and-bound algorithms [19, 40].

In this work, we thus aim for a 'neural boost' of DP algorithms, by using a graph neural network for scoring partial solutions. Prior work on 'neural' vehicle routing has focused on auto-regressive models [57, 5, 13, 29], but they have high computational cost when combined with (any form of) search, as the model needs to be evaluated for each partial solution considered. Instead, we use (for TSP) and adapt (for VRP and TSPTW) a model to predict a heatmap indicating promising edges [26], and define the *score* of a partial solution as the 'heat' of the edges it contains (plus an estimate of the 'heat-to-go' or *potential* of the solution). As the neural network only needs to be evaluated *once* for each instance, this enables a *much larger search* (defined by $B$), making a good trade-off between quality and computational cost. Additionally, we can apply a threshold to the heatmap to define a sparse graph on which to run the DP algorithm, reducing the runtime by eliminating many solutions.

Figure 2 illustrates the overall DPDP algorithm. In Section 4, we show that DPDP significantly improves over 'classic' restricted DP algorithms (with the same $B$). Additionally, we show that DPDP outperformes most other 'neural' approaches for TSP, VRP and TSPTW and is competitive with the highly-optimized LKH solver [24] for VRP, while achieving similar results much faster for TSP and TSPTW. For TSPTW, DPDP also outperforms the best open-source solver we could find [10], illustrating the power of DPDP to handle difficult hard constraints (time windows).

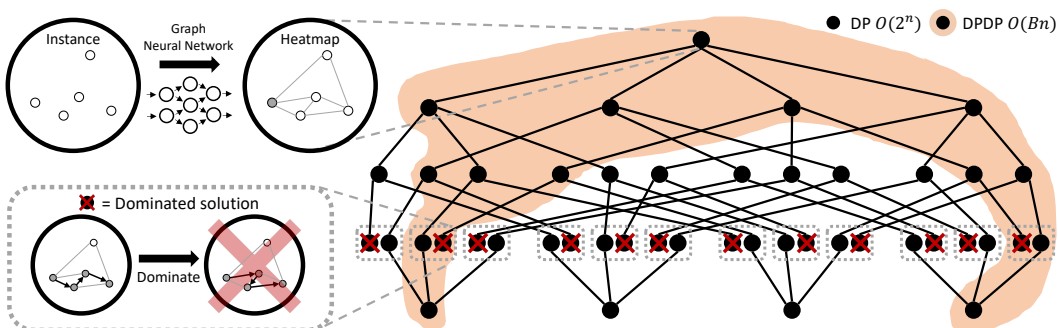

Figure 2: Deep Policy Dynamic Programming for the TSP. A GNN creates a (sparse) heatmap indicating promising edges, after which a tour is constructed using forward dynamic programming. In each step, at most $B$ solutions are expanded according to the heatmap policy, restricting the size of the search space. Partial solutions are dominated by shorter (lower cost) solutions with the same DP state: the same nodes visited (marked grey) and current node (indicated by dashed rectangles).

## 2 Related work

DP has a long history as an exact solution method for routing problems [31, 50], e.g. for the TSP with time windows [15] and precedence constraints [39], but typically limited to small problems only, due to the curse of dimensionality. Restricted DP (with heuristic policies) has been used to address, e.g., the time dependent TSP [37], and has been generalized into a flexible framework for VRPs with different types of practical constraints [20]. DP approaches have also been shown to be useful in settings with difficult practical issues such as time-dependent travel times and driving regulations [28] or stochastic demands [42]. For a thorough investigation of modelling choices of DP for routing (and scheduling), see [53]. For sparse graphs, alternative, but less flexible, formulations can be used [8].

Despite the flexibility, constructive DP methods have not gained much popularity compared to heuristic search approaches such as Ruin and Recreate [47], Adaptive Large Neighborhood Search [46], LKH [24] or FILO [1]. While highly effective, these methods are limited in their flexibility as special operators need to be engineered for different types of problems. While restricted DP was shown to have superior performance on *realistic* VRPs with many constraints [20], the performance gap of around 10% for standard (benchmark) VRPs (with time windows) is too large to popularize the restricted dynamic programming approach. We argue that the missing ingredient for restricted dynamic programming is the availability of a strong but computationally cheap policy for selecting which solutions should be considered, which is the motivation behind DPDP.

In the machine learning community, recent advances have significantly improved deep neural networks (DNNs) to perform tasks such as image classification and machine translation [32]. After the first deep learning model was trained (using example solutions) to construct TSP tours [57], many improvements have been proposed, e.g. different training strategies such as reinforcement learning (RL) [5, 27, 12, 30] and model architectures, which enabled the same idea to be used for other routing problems [41, 29, 13, 45, 16, 60]. Most constructive neural methods are *auto-regressive*, evaluating the model many times to predict one node at the time, but other works have considered predicting a 'heatmap' of promising edges *at once* [43, 26, 17], which allows a tour to be constructed (using sampling or beam search) without further evaluating the model. An alternative direction is 'learning to search', where a neural network is used to guide a search procedure such as local search [7, 35, 18, 59, 25]. Some works have attempted scaling to larger instances beyond 100 nodes, which remains challenging [36, 17]. The combination of machine learning with DP has been proposed in limited settings [62, 52, 61]. Most related to our approach, a DP algorithm for TSPTW, guided by an RL agent, was implemented using an existing solver [6] and a neural network predicting edges has been combined with tree search [34] and local search for maximum independent set (MIS). For a wider view on machine learning for routing problems and combinatorial optimization, see [38, 54].

## 3 Deep Policy Dynamic Programming

DPDP uses an existing graph neural network [26] (which we modify for VRP and TSPTW) to predict a heatmap of promising edges, which is used to derive the policy for scoring partial solutions in the DP algorithm. The DP algorithm starts with a *beam* of a single initial (empty) solution. It then proceeds by iterating the following steps: (1) all solutions on the beam are expanded, (2) dominated solutions are removed for each *DP state*, (3) the $B$ best solutions according to the scoring policy define the beam for the next iteration. These steps are illustrated in Fig. 2. The objective function is used to select the best solution from the final beam. The resulting algorithm is a *beam search* over the *DP state space* (which is *not* a 'standard beam search' over the *solution space*!) and we call $B$ the *beam size*. DPDP is asymptotically optimal as using $B = n \cdot 2^n$ for a TSP with $n$ nodes guarantees optimal results, but choosing smaller $B$ allows to trade performance for computational cost.

DPDP is a generic framework that can be applied to different problems, by defining the following ingredients: (1) the **state variables** to track while constructing solutions, (2) the **initial solution**, (3) **feasible actions** to expand solutions, (4) rules to define **dominated solutions** and (5) a **scoring policy** for selecting the $B$ solutions to keep. A solution is always (uniquely) defined as a sequence of actions, which allows the DP algorithm to construct the final solution by backtracking. In the next sections, we define these ingredients for the TSP, VRP and TSPTW.

 ## 3.1 Travelling Salesman Problem

We implement DPDP for Euclidean TSPs with $n$ nodes on a (sparse) graph, where the cost for edge $(i, j)$ is given by $c_{ij}$, the Euclidean distance between the coordinates of nodes $i$ and $j$.

For each partial solution, defined by a sequence of actions $\boldsymbol{a}$, the **state variables** are $\mathrm{cost}(\boldsymbol{a})$, the total *cost* (distance), $\mathrm{current}(\boldsymbol{a})$, the current node, and $\mathrm{visited}(\boldsymbol{a})$, the set of visited nodes (including the start node). Without loss of generality, we let 0 be the start node, so we initialize the beam at step $t = 0$ with the empty **initial solution** with $\mathrm{cost}(\boldsymbol{a}) = 0$, $\mathrm{current}(\boldsymbol{a}) = 0$ and $\mathrm{visited}(\boldsymbol{a}) = \{0\}$. At step $t$, the action $a_t \in \{0, ..., n-1\}$ indicates the next node to visit, and is a **feasible action** for a partial solution $\boldsymbol{a} = (a_0, ..., a_{t-1})$ if $(a_{t-1}, a_t)$ is an edge in the graph and $a_t \notin \mathrm{visited}(\boldsymbol{a})$, or, when all are visited, if $a_t = 0$ to return to the start node. When expanding the solution to $\boldsymbol{a}' = (a_0, ..., a_t)$, we can compute the state variables incrementally as:

$$\mathrm{cost}(\boldsymbol{a}') = \mathrm{cost}(\boldsymbol{a}) + c_{\mathrm{current}(\boldsymbol{a}),a_t}, \quad \mathrm{current}(\boldsymbol{a}') = a_t, \quad \mathrm{visited}(\boldsymbol{a}') = \mathrm{visited}(\boldsymbol{a}) \cup \{a_t\}. \quad (1)$$

A (partial) solution $\boldsymbol{a}$ is a **dominated solution** if there exists a (dominating) solution $\boldsymbol{a}^*$ such that $\mathrm{visited}(\boldsymbol{a}^*) = \mathrm{visited}(\boldsymbol{a})$, $\mathrm{current}(\boldsymbol{a}^*) = \mathrm{current}(\boldsymbol{a})$ and $\mathrm{cost}(\boldsymbol{a}^*) < \mathrm{cost}(\boldsymbol{a})$. The tuple $(\mathrm{visited}(\boldsymbol{a}), \mathrm{current}(\boldsymbol{a}))$ we refer to as the *DP state*, so removing all dominated partial solutions, we keep exactly one minimum-cost solution for each unique DP state[1]. Since a solution can only dominate other solutions with the same set of visited nodes, we only need to remove dominated solutions from sets of solutions with the same number of actions. Therefore, we can efficiently execute the DP algorithm in iterations, where at step $t$ all solutions have (after $t$ actions) $t + 1$ visited nodes (including the start node), keeping the memory need at $O(B)$ states (with $B$ the beam size).

We define the **scoring policy** using a pretrained model [26], which takes as input node coordinates and edge distances to predict a raw 'heatmap' value $\hat{h}_{ij} \in (0, 1)$ for each edge $(i, j)$. The model was trained to predict optimal solutions, so $\hat{h}_{ij}$ can be seen as the probability that edge $(i, j)$ is in the optimal tour. We force the heatmap to be symmetric thus we define $h_{ij} = \max\{\hat{h}_{ij}, \hat{h}_{ji}\}$. The policy is defined using the heatmap values, in such a way to select the (partial) solutions with the largest total *heat*, while also taking into account the (heat) *potential* for the unvisited nodes. The policy thus selects the $B$ solutions which have the highest *score*, defined as $\mathrm{score}(\boldsymbol{a}) = \mathrm{heat}(\boldsymbol{a}) + \mathrm{potential}(\boldsymbol{a})$, with $\mathrm{heat}(\boldsymbol{a}) = \sum_{i=1}^{t-1} h_{a_{i-1}, a_i}$, i.e. the sum of the heat of the edges, which can be computed incrementally when expanding a solution. The potential is added as an estimate of the 'heat-to-go' (similar to the heuristic in $A^*$ search) for the remaining nodes, and avoids the 'greedy pitfall' of selecting the best edges while skipping over nearby nodes, which would prevent good edges from being used later. It is defined as $\mathrm{potential}(\boldsymbol{a}) = \mathrm{potential}_0(\boldsymbol{a}) + \sum_{i \notin \mathrm{visited}(\boldsymbol{a})} \mathrm{potential}_i(\boldsymbol{a})$ with $\mathrm{potential}_i(\boldsymbol{a}) = w_i \sum_{j \notin \mathrm{visited}(\boldsymbol{a})} \frac{h_{ji}}{\sum_{k=0}^{n-1} h_{ki}}$, where $w_i$ is the node *potential weight* given by $w_i = (\max_j h_{ji}) \cdot (1 - 0.1(\frac{c_{i0}}{\max_j c_{j0}} - 0.5))$. By normalizing the heatmap values for incoming edges, the (remaining) potential for node $i$ is initially equal to $w_i$ but decreases as good edges become infeasible due to neighbours being visited. The node potential weight $w_i$ is equal to the maximum incoming edge heatmap value (an upper bound to the heat contributed by node $i$), which gets multiplied by a factor 0.95 to 1.05 to give a higher weight to nodes closer to the start node, which we found helps to encourage the algorithm to keep edges that enable to return to the start node. The overall heat + potential function identifies promising partial solutions and is computationally cheap.

## 3.2 Vehicle Routing Problem

For the VRP, we add a special depot node to the graph, indicated by DEP. Each node $i$ has a demand $d_i$, and the goal is to find multiple routes, which have a limited capacity denoted by CAPACITY.

Additionally to the TSP **state variables** $\mathrm{cost}(\boldsymbol{a})$, $\mathrm{current}(\boldsymbol{a})$ and $\mathrm{visited}(\boldsymbol{a})$, we keep track of $\mathrm{capacity}(\boldsymbol{a})$, which is the *remaining* capacity in the current route/vehicle. A solution starts at the depot, so we initialize the beam at step $t = 0$ with the empty **initial solution** with $\mathrm{cost}(\boldsymbol{a}) = 0$, $\mathrm{current}(\boldsymbol{a}) = \mathrm{DEP}$, $\mathrm{visited}(\boldsymbol{a}) = \emptyset$ and $\mathrm{capacity}(\boldsymbol{a}) = \mathrm{CAPACITY}$. For the VRP, we do not consider visiting the depot as a separate action. Instead, we define $2n$ actions, where $a_t \in \{0, ..., 2n-1\}$. The actions $0, ..., n-1$ indicate a *direct* move from the current node to node $a_t$, whereas the actions

---

[1]If we have multiple partial solutions with the same state and cost, we can arbitrarily choose one to dominate the other(s), for example the one with the lowest index of the current node.

$n, ..., 2n - 1$ indicate a move to node $a_t - n$ *via the depot*. **Feasible actions** are those that move to unvisited nodes via edges in the graph and obey the following constraints. For the first action $a_0$ there is no choice and we constrain (for convenience of implementation) $a_0 \in \{n, ..., 2n - 1\}$. A direct move ($a_t < n$) is only feasible if $d_{a_t} \leq \text{capacity}(\boldsymbol{a})$ and updates the state similar to TSP but reduces remaining capacity by $d_{a_t}$. A move via the depot is always feasible (respecting the graph edges and assuming $d_i \leq \text{CAPACITY } \forall i$) as it resets the vehicle CAPACITY before subtracting demand, but incurs the 'via-depot cost' $c_{ij}^{\text{DEP}} = c_{i,\text{DEP}} + c_{\text{DEP},j}$. When all nodes are visited, we allow a special action to return to the depot. This somewhat unusual way of representing a CVRP solution has desirable properties similar to the TSP formulation: at step $t$ we have exactly $t$ nodes visited, and we can run the DP in iterations, removing dominated solutions at each step $t$.

For VRP, a partial solution $\boldsymbol{a}$ is a **dominated solution** dominated by $\boldsymbol{a}^*$ if $\text{visited}(\boldsymbol{a}^*) = \text{visited}(\boldsymbol{a})$ and $\text{current}(\boldsymbol{a}^*) = \text{current}(\boldsymbol{a})$ (i.e. $\boldsymbol{a}^*$ corresponds to the same DP state) and $\text{cost}(\boldsymbol{a}^*) \leq \text{cost}(\boldsymbol{a})$ and $\text{capacity}(\boldsymbol{a}^*) \geq \text{capacity}(\boldsymbol{a})$, with *at least one of the two inequalities being strict*. This means that for each DP state, given by the set of visited nodes and the current node, we do not only keep the (single) solution with lowest cost (as in the TSP algorithm), but keep the complete set of pareto-efficient solutions in terms of cost and remaining vehicle capacity. This is because a higher cost partial solution may still be preferred if it has more remaining vehicle capacity, and vice versa.

For the VRP **scoring policy**, we modify the model [26] to include the depot node and demands. The special depot node gets a separate learned initial embedding parameter, and we add additional edge types for connections to the depot, to mark the depot as being special. Additionally, each node gets an extra input (next to its coordinates) corresponding to $d_i/\text{CAPACITY}$ (where we set $d_{\text{DEP}} = 0$). Apart from this, the model remains exactly the same[2]. The model is trained on example solutions from LKH [24] (see Section 4.2), which are not optimal, but still provide a useful training signal. Compared to TSP, the definition of the heat is slightly changed to accommodate for the 'via-depot actions' and is best defined incrementally using the 'via-depot heat' $h_{ij}^{\text{DEP}} = h_{i,\text{DEP}} \cdot h_{\text{DEP},j} \cdot 0.1$, where multiplication is used to keep heat values interpretable as probabilities and in the range $(0, 1)$. The additional penalty factor of 0.1 for visiting the depot encourages the algorithm to minimize the number of vehicles/routes. The initial heat is 0 and when expanding a solution $\boldsymbol{a}$ to $\boldsymbol{a}'$ using action $a_t$, the heat is incremented with either $h_{\text{current}(\boldsymbol{a}),a_t}$ (if $a_t < n$) or $h_{\text{current}(\boldsymbol{a}),a_t-n}^{\text{DEP}}$ (if $a_t \geq n$). The potential is defined similarly to TSP, replacing the start node 0 by DEP.

## 3.3 Travelling Salesman Problem with Time Windows

For the TSPTW, we also have a special depot / start node 0, and each node $i$ has a time window defined by $(l_i, u_i)$ in which the node should be visited, assuming travel time is equal to cost/distance. It is allowed to wait if arrival at a node is before $l_i$, but arrival cannot be after $u_i$ (i.e. the constraint is hard). We consider the objective to be minimizing total *cost*, but minimizing total time (or *makespan*) only requires training on different example solutions. Due to the hard constraints, TSPTW is typically considered more challenging to solve than plain TSP, for which every solution is feasible.

The **state variables** and **initial solution** are equal to TSP except that we add $\text{time}(\boldsymbol{a})$ which is initially 0 ($= l_0$). **Feasible actions** $a_t \in \{0, ..., n - 1\}$ are those that move to unvisited nodes via edges in the graph such that the arrival time is no later than $u_{a_t}$ and do not directly eliminate the possibility to visit other nodes in time[3]. Expanding a solution $\boldsymbol{a}$ to $\boldsymbol{a}'$ using action $a_t$ updates the time as $\text{time}(\boldsymbol{a}') = \max\{\text{time}(\boldsymbol{a}) + c_{\text{current}(a),a_t}, l_{a_t}\}$.

For each DP state, we keep all efficient solutions in terms of cost and time, so a partial solution $\boldsymbol{a}$ is a **dominated solution** dominated by $\boldsymbol{a}^*$ if $\boldsymbol{a}^*$ has the same DP state ($\text{visited}(\boldsymbol{a}^*) = \text{visited}(\boldsymbol{a})$ and $\text{current}(\boldsymbol{a}^*) = \text{current}(\boldsymbol{a})$) and is strictly better in terms of cost and time, i.e. $\text{cost}(\boldsymbol{a}^*) \leq \text{cost}(\boldsymbol{a})$ and $\text{time}(\boldsymbol{a}^*) \leq \text{time}(\boldsymbol{a})$, with *at least one of the two inequalities being strict*.

The model [26] for the **scoring policy** is adapted to include the time windows $(l_i, u_i)$ as node features (in the same unit as coordinates and costs), and we use a special embedding for the depot similar to VRP. Due to the time dimension, a TSPTW solution is *directed*, and edge $(i, j)$ may be good whereas $(j, i)$ may be not, so we adapt the model to enable predictions $h_{ij} \neq h_{ji}$ (see details in Appendix B). We generated example training solutions using (heuristic) DP with a large beam size, which was faster than using LKH. Given the heat predictions, the score (heat + potential) is exactly as for TSP.

---

[2]Except that we do not use the K-nearest neighbour feature [26] as it contains no additional information.

[3]E.g., arriving at a node $i$ at $t = 10$ (including waiting) is not feasible if node $j$ has $u_j = 12$ and $c_{ij} = 3$.

## 3.4 Graph sparsity

As described, the DP algorithm can take into account a sparse graph to define feasible expansions. As our problems are defined on sets of nodes rather than graphs, the use of a sparse graph is an artificial design choice, which allows to significantly reduce the runtime but may sacrifice the possibility to find good or optimal tours. We propose two different strategies for defining the sparse graph on which to run the DP: thresholding the heatmap values $h_{ij}$ and using the K-nearest neighbour (KNN) graph. By default, we use a (low) heatmap threshold of $10^{-5}$, which rules out most of the edges as the model confidently predicts (close to) 0 for most edges. This is a secondary way to leverage the neural network (independent of the scoring policy), which can be seen as a form of learned *problem reduction* [49]. For symmetric problems (TSP and VRP), we add KNN edges in both directions. For the VRP, we additionally connect each node to the depot (and vice versa) to ensure feasibility.

## 3.5 Implementation & hyperparameters

We implement DPDP using PyTorch [44] to leverage batched computation on the GPU. For details, see Appendix A. Our code is publicly available.[4] DPDP has very few hyperparameters, but the heatmap threshold of $10^{-5}$ and details like the functional form of e.g. the scoring policy are 'educated guesses' or manually tuned on a few validation instances and can likely be improved. The runtime is influenced by implementation choices which were manually selected using a few validation instances.

# 4 Experiments

## 4.1 Travelling Salesman Problem

In Table 1 we report our main results for DPDP with beam sizes of 10K (10 thousand) and 100K, for the TSP with 100 nodes on a commonly used test set [29]. We report results using Concorde [2], LKH [24] and Gurobi [22], as well as recent results of the strongest methods using neural networks ('neural approaches') from literature. Running times for solving 10000 instances *after training* should be taken as rough indications as some are on different machines, typically with 1 GPU or a many-core CPU (8 - 32). The costs indicated with * are not directly comparable due to slight dataset differences [17]. Times for generating heatmaps (if applicable) is reported separately (as the first term) from the running time for MCTS [17] or DP. DPDP achieves close to optimal results, strictly outperforming the neural baselines achieving better results in less time (except POMO [30], see Section 4.2).

---

[4]`https://github.com/?????`, to be disclosed after review

Table 1: Mean cost, gap and *total time* to solve 10000 TSP/VRP instances *after training*.

| PROBLEM METHOD | TSP100 COST | GAP | TIME | VRP100 COST | GAP | TIME |
|---|---|---|---|---|---|---|
| CONCORDE [2] | 7.765 | 0.000 % | 6M | | | |
| HYBRID GENETIC SEARCH [56, 55] | | | | 15.563 | 0.000 % | 6H11M |
| GUROBI [22] | 7.776 | 0.151 % | 31M | | | |
| LKH [24] | 7.765 | 0.000 % | 42M | 15.647 | 0.536 % | 12H57M |
| GNN HEATMAP + BEAM SEARCH [26] | 7.87 | 1.39 % | 40M | | | |
| LEARNING 2-OPT HEURISTICS [9] | 7.83 | 0.87 % | 41M | | | |
| MERGED GNN HEATMAP + MCTS [17] | 7.764* | 0.04 % | 4M + 11M | | | |
| ATTENTION MODEL + SAMPLING [29] | 7.94 | 2.26 % | 1H | 16.23 | 4.28 % | 2H |
| STEP-WISE ATTENTION MODEL [60] | 8.01 | 3.20 % | 29S | 16.49 | 5.96 % | 39S |
| LEARNING IMPROV. HEURISTICS [59] | 7.87 | 1.42 % | 2H | 16.03 | 3.00 % | 5H |
| ATTENTION MODEL + POMO [30] | 7.77 | 0.14 % | 1M | 15.76 | 1.26 % | 2M |
| NEUREWRITER [7] | | | | 16.10 | 3.45 % | 1H |
| DYNAMIC ATTN. MODEL + 2-OPT [45] | | | | 16.27 | 4.54 % | 6H |
| NEUR. LRG. NEIGHB. SEARCH [25] | | | | 15.99 | 2.74 % | 1H |
| LEARN TO IMPROVE [35] | | | | 15.57* | - | 4000H |
| DPDP 10K | 7.765 | 0.009 % | 10M + 16M | 15.830 | 1.713 % | 10M + 50M |
| DPDP 100K | 7.765 | 0.004 % | 10M + 2H35M | 15.694 | 0.843 % | 10M + 5H48M |
| DPDP 1M | | | | 15.627 | 0.409 % | 10M + 48H27M |

Table 2: Mean cost, gap and *total time* to solve 10000 realistic [51] VRP100 instances after training.

| METHOD | COST | GAP | TIME (1 GPU OR 16 CPUs) | TIME (4 GPUs OR 32 CPUs) |
|---|---|---|---|---|
| HGS [56, 55] | 18050 | 0.000 % | 7H53M | 3H56M |
| LKH [24] | 18133 | 0.507 % | 25H32M | 12H46M |
| DPDP 10K | 18414 | 2.018 % | 10M + 50M | 2M + 13M |
| DPDP 100K | 18253 | 1.127 % | 10M + 5H48M | 2M + 1H27M |
| DPDP 1M | 18168 | 0.659 % | 10M + 48H27M | 2M + 12H7M |

## 4.2 Vehicle Routing Problem

For the VRP, we train the model using 1 million instances of 100 nodes, generated according to the distribution described by [41] and solved using one run of LKH [24]. We train using a batch size of 48 and a learning rate of $10^{-3}$ (selected as the result of manual trials to best use our GPUs), for (at most) 1500 epochs of 500 training steps (following [26]) from which we select the saved checkpoint with the lowest validation loss. We use the validation and test sets by [29].

Table 1 shows the results compared to a recent implementation of Hybrid Genetic Search (HGS)[5], a SOTA heuristic VRP solver [56, 55]. HGS is faster and improves around 0.5% over LKH, which is typically considered the baseline in related work. We present the results for LKH, as well as the strongest neural approaches and DPDP with beam sizes up to 1 million. Some results used 2000 (different) instances [35] and cannot be directly compared[6]. DPDP outperforms all other neural baselines, except POMO [30], which delivers good results very quickly by exploiting symmetries in the problem. However, as it cannot (easily) improve further with additional runtime, we consider this contribution orthogonal to DPDP. DPDP is competitive to LKH (see also Section 4.4).

**More realistic instances** We also train the model and run experiments with instances with 100 nodes from a more realistic and challenging data distribution [51]. This distribution, commonly used in the routing community, has greater variability, in terms of node clustering and demand distributions. LKH failed to solve two of the test instances, which we found out is because LKH by default uses a fixed number of routes equal to a lower bound, given by $\left\lceil \frac{\sum_{i=0}^{n-1} d_i}{\text{CAPACITY}} \right\rceil$, which may be infeasible[7]. Therefore we solve these instances by rerunning LKH with an unlimited number of allowed routes (which in general gives worse results, see Section 4.4).

DPDP was run on a machine with 4 GPUs, but we also report (estimated) runtimes for 1 GPU (1080Ti), and we compare against 16 or 32 CPUs for HGS and LKH. In Table 2 it can be seen that the difference with LKH is, as expected, slightly larger than for the simpler dataset, but still below 1% for beam sizes of 100K-1M. We also observed a higher validation loss, so it may be possible to improve results using more training data. Nevertheless, finding solutions within 1% of the specialized SOTA HGS algorithm, and even closer to LKH, is impressive for these challenging instances, and we consider the runtime (for solving 10K instances) acceptable, especially when using multiple GPUs.

## 4.3 TSP with Time Windows

For the TSP with hard time window constraints, we use the data distribution by [6] and use their set of 100 test instances with 100 nodes. These were generated with small time windows, resulting in a small feasible search space, such that even with very small beam sizes, our DP implementation solves these instances optimally, eliminating the need for a policy. Therefore, we also consider a more difficult distribution similar to [10], which has larger time windows which are more difficult as the feasible search space is larger[8] [15]. For details, see Appendix B. For both distributions, we generate training data and train the model exactly as we did for the VRP.

---

[5] https://github.com/vidalt/HGS-CVRP
[6] The running time of 4000 hours (167 days) for 10K instances is estimated from 24min/instance [35].
[7] For example, three nodes with a demand of two cannot be assigned to two routes with a capacity of three.
[8] Up to a limit, as making the time windows infinite size reduces the problem to plain TSP.

Table 3: Mean cost, gap and *total time* to solve TSPTW100 instances after training.

| PROBLEM METHOD | SMALL TIME WINDOWS [6] (100 INST.) | | | | LARGE TIME WINDOWS [10] (10K INST.) | | | |
|---|---|---|---|---|---|---|---|---|
| | COST | GAP | FAIL | TIME | COST | GAP | FAIL | TIME |
| GVNS 30x [10] | 5129.58 | 0.000 % | | 7S | 2432.112 | 0.000 % | | 37M15S |
| GVNS 1x [10] | 5129.58 | 0.000 % | | <1S | 2457.974 | 1.063 % | | 1M4S |
| LKH 1x [24] | 5130.32 | 0.014 % | 1.00 % | 5M48S | 2431.404 | -0.029 % | | 34H58M |
| BAB-DQN* [6] | 5130.51 | 0.018 % | | 25H | | | | |
| ILDS-DQN* [6] | 5130.45 | 0.017 % | | 25H | | | | |
| DPDP 10K | 5129.58 | 0.000 % | | 6S + 1S | 2431.143 | -0.040 % | | 10M + 8M7S |
| DPDP 100K | 5129.58 | 0.000 % | | 6S + 1S | 2430.880 | - 0.051 % | | 10M + 1H16M |

Table 3 shows the results for both data distributions, which are reported in terms of the difference to General Variable Neighbourhood Search (GVNS) [10], the best open-source solver for TSPTW we could find[9], using 30 runs. For the small time window setting, both GVNS and DPDP find optimal solutions for all 100 instances in just 7 seconds (in total, either on 16 CPUs or a single GPU). LKH fails to solve one instance, but finds close to optimal solutions, but around 50 times slower. BaB-DQN* and ILDS-DQN* [6], methods combining an existing solver with an RL trained neural policy, take around 15 minutes *per instance* (orders of magnitudes slower) to solve most instances to optimality. Due to complex set-up, we were unable to run BaB-DQN* and ILDS-DQN* ourselves for the setting with larger time windows. In this setting, we find DPDP outperforms both LKH (where DPDP is orders of magnitude faster) and GVNS, in both speed and solution quality. This illustrates that DPDP, due to its nature, is especially well suited to handle constrained problems.

## 4.4 Ablations

**Scoring policy** To evaluate the value of different components of DPDP's **GNN Heat + Potential** scoring policy, we compare against other variants. **GNN Heat** is the version without the potential, whereas **Cost Heat + Potential** and **Cost Heat** are variants that use a 'heuristic' $\hat{h}_{ij} = \frac{c_{ij}}{\max_k c_{ik}}$ instead of the GNN. **Cost** directly uses the current cost of the solution, and can be seen as 'classic' restricted DP. Finally, **BS GNN Heat + Potential** uses beam search without dynamic programming, i.e. without removing dominated solutions. To evaluate only the scoring policy, each variant uses the fully connected graph (knn = $n - 1$). Figure 3a shows the value of DPDP's potential function, although even without it results are still significantly better than 'classic' heuristic DP variants using cost-based scoring policies. Also, it is clear that using DP significantly improves over a standard beam search (by removing dominated solutions). Lastly, the figure illustrates how the time for generating the heatmap using the neural network, despite its significant value, only makes up a small portion of the total runtime.

---

[9]https://github.com/sashakh/TSPTW

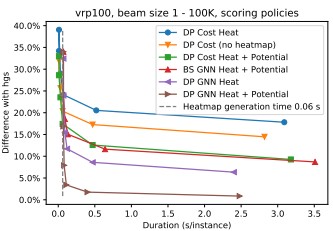
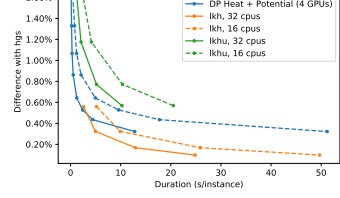
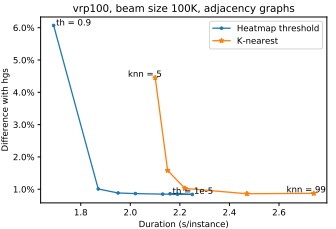

(a) Different scoring policies, as well as 'pure' beam search, for beam sizes 1, 10, 100, 1000, 10K, 100K.

(b) Beam sizes 10K, 25K, 50K, 100K, 250K, 500K, 1M, 2.5M compared against LKH(U) with 1, 2, 5 and 10 runs.

(c) Sparsities with heatmap thresholds 0.9, 0.5, 0.2, 0.1, $10^{-2}$, $10^{-3}$, $10^{-4}$, $10^{-5}$ and knn = 5, 10, 20, 50, 99. Beam size 100K.

Figure 3: DPDP ablations on 100 validation instances of VRP with 100 nodes.

**Beam size** DPDP allows to trade off the performance vs. the runtime using the beam size $B$ (and to some extent the graph sparsity, see Section 4.4). We illustrate this trade-off in Figure 3b, where we evaluate DPDP on 100 validation instances for VRP, with different beam sizes from 10K to 2.5M. We also report the trade-off curve for the LKH(U), which is the strongest baseline that can also solve different problems. We vary the runtime using 1, 2, 5 and 10 runs (returning the best solution). LKHU(nlimited) is the version which allows an unlimited number of routes (see Section 4.2). It is hard to compare GPU vs CPU, so we report (estimated) runtimes for different hardware, i.e. 1 or 4 GPUs (with 3 CPUs per GPU) and 16 or 32 CPUs. We report the difference (i.e. the gap) with HGS, analog to how results are reported in Table 1. We emphasize that in most related work (e.g. [29]), the strongest baseline considered is one run of LKH, so we compare against a much stronger baseline. Also, our goal is not to outperform HGS (which is SOTA and specific to VRP) or LKH, but to show DPDP has reasonable performance, while being a flexible framework for other (routing) problems.

**Graph sparsity** We test the two graph sparsification strategies described in Section 3.4 as another way to trade off performance and runtime of DPDP. In Figure 3c, we experiment with different heatmap thresholds from $10^{-5}$ to 0.9 and different values for KNN from 5 to 99 (fully connected). The heatmap threshold strategy clearly outperforms the KNN strategy as it yields the same results using sparser graphs (and lower runtimes). This illustrates that the heatmap threshold strategy is more informed than the KNN strategy, confirming the value of the neural network predictions.

# 5 Discussion

In this paper we introduced Deep Policy Dynamic Programming, which combines machine learning and dynamic programming for solving vehicle routing problems. The method yields close to optimal results for TSPs with 100 nodes and is competitive to the highly optimized LKH [24] solver for VRPs with 100 nodes. On the TSP with time windows, DPDP also outperforms LKH, being significanlty faster, as well as GVNS [10], the best open source solver we could find. Given that DPDP was not specifically designed for TSPTW, and still has possibilities for improvement, we consider this an impressive and promising achievement.

The constructive nature of DPDP (combined with search) allows to efficiently address hard constraints such as time windows, which are typically considered challenging in neural combinatorial optimization [5, 29] and are also difficult for local search heuristics (as they need to maintain feasibility while adapting a solution). Given our results on TSP, VRP and TSPTW, and the flexibility of DP as a framework, we think DPDP has great potential for solving many more variants of routing problems, and possibly even other problems that can be formulated using DP (e.g. job shop scheduling [21]). We hope that our work brings machine learning research for combinatorial optimization closer to the operations research (especially vehicle routing) community, by combining machine learning with DP and evaluating the resulting new framework on different data distributions used by different communities [41, 51, 6, 10].

**Scope, limitations & future work** Deep learning for combinatorial optimization is a recent research direction, which could significantly impact the way practical optimization problems get solved in the future. Currently, however, it is still hard to beat most SOTA problem specific solvers from the OR community. Despite our success for TSPTW, DPDP is not yet a practical alternative in general, but we do consider our results as highly encouraging for further research. We belief such research could yield significant further improvement by addressing key current limitations: (1) the scalability to larger instances, (2) the dependency on example solutions and (3) the heuristic nature of the scoring function. First, while 100 nodes is not far from the size of common benchmarks (100 - 1000 for VRP [51] and 20 - 200 for TSPTW [10]), scaling is a challenge, mainly due to the 'fully-connected' $O(n^2)$ graph neural network. Future work could reduce this complexity following e.g. [33]. The dependency on example solutions from an existing solver also becomes more prominent for larger instances, but could potentially be removed by 'bootstrapping' using DP itself as we, in some sense, have done for TSPTW (see Section 3.3). Future work could iterate this process to train the model 'tabula rasa' (without example solutions), where DP could be seen analogous to MCTS in *AlphaZero* [48]. Lastly, the heat + potential score function is a well-motivated but heuristic function that was manually designed as a function of the predicted heatmap. While it worked well for the three problems we considered, it may need suitable adaption for other problems. Training this function end-to-end [11, 58], while keeping a low computational footprint, would be an interesting topic for future work.

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
