## A Implementation

We implement the dynamic programming algorithm on the GPU using PyTorch [44]. While mostly used as a Deep Learning framework, it can be used to speed up generic (vectorized) computations.

### A.1 Beam variables

For each solution in the beam, we keep track of the following variables (storing them for all solutions in the beam as a vector): the cost, current node, visited nodes and (for VRP) the remaining capacity or (for TSPTW) the current time. As explained, these variables can be computed incrementally when generating expansions. Additionally, we keep a variable vector *parent*, which, for each solution in the current beam, tracks the index of the solution in the previous beam that generated the expanded solution. To compute the score of the policy for expansions efficiently, we also keep track of the score for each solution and the potential for each node for each solution incrementally.

We do not keep past beams in memory, but at the end of each iteration, we store the vectors containing the parents as well as last actions for each solution on the *trace*. As the solution is completely defined by the sequence of actions, this allows to backtrack the solution after the algorithm has finished. To save GPU memory (especially for larger beam sizes), we store the $O(Bn)$ sized trace on the CPU memory.

For efficiency, we keep the set of visited nodes as a bitmask, packed into 64-bit long integers (2 for 100 nodes). Using bitwise operations with the packed adjacency matrix, this allows to quickly check feasible expansions (but we need to *unpack* the mask into boolean vectors to find all feasible expansions explicitly). Figure 4a shows an example of the beam (with variables related to the policy and backtracking omitted) for the VRP.

### A.2 Generating non-dominated expansions

A solution $a$ can only dominate a solution $a'$ if visited($a$) = visited($a'$) and current($a$) = current($a'$), i.e. if they correspond to the same *DP state*. If this is the case, then, if we denote by parent($a$) the parent solution from which $a$ was expanded, it holds that

$$\text{visited(parent}(a)) = \text{visited}(a) \setminus \{\text{current}(a)\}$$
$$= \text{visited}(a') \setminus \{\text{current}(a')\}$$
$$= \text{visited(parent}(a')).$$

This means that only expansions from solutions with the same set of visited nodes can dominate each other, so we only need to check for dominated solutions among groups of expansions originating from parent solutions with the same set of visited nodes. Therefore, before generating the expansions, we group the current beam (the parents of the expansions) by the set of visited nodes (see Figure 4a). This can be done efficiently, e.g. using a lexicographic sort of the packed bitmask representing the sets of visited nodes[10].

### A.2.1 Travelling Salesman Problem

For TSP, we can generate (using boolean operations) the $B \times n$ matrix with boolean entries indicating feasible expansions (with $n$ action columns corresponding to $n$ nodes, similar to the $B \times 2n$ matrix for VRP in Figure 4a), i.e. nodes that are unvisited and adjacent to the current node. If we find positive entries sequentially for each column (e.g. by calling TORCH.NONZERO on the transposed matrix), we get all expansions grouped by the combination of action (new current node) and parent set of visited nodes, i.e. grouped by the DP state. We can then trivially find the segments of consecutive expansions corresponding to the same DP state, and we can efficiently find the minimum cost solution for each segment, e.g. using TORCH_SCATTER [11].

---

[10]For efficiency, we use a custom function similar to TORCH.UNIQUE, and argsort the returned inverse after which the resulting permutation is applied to all variables in the beam.

[11]https://github.com/rusty1s/pytorch_scatter

| Cost | Capacity | Visited | Current | Direct 0 | 1 | 2 | 3 | 4 | Via-depot 0 | 1 | 2 | 3 | 4 |
|---|---|---|---|---|---|---|---|---|---|---|---|---|---|
| 10 | 5 | 01101 | 1 | 1 | 0 | 0 | 0 | 0 | 1 | 0 | 0 | 1 | 0 |
| 12 | 8 | 01101 | 1 | 1 | 0 | 0 | 1 | 0 | 1 | 0 | 0 | 1 | 0 |
| 13 | 7 | 01101 | 2 | 1 | 0 | 0 | 1 | 0 | 0 | 0 | 0 | 0 | 0 |
| 8 | 3 | 01101 | 4 | 0 | 0 | 0 | 0 | 0 | 1 | 0 | 0 | 1 | 0 |
| 11 | 7 | 10101 | 0 | 0 | 1 | 0 | 1 | 0 | 0 | 0 | 0 | 1 | 0 |
| 12 | 6 | 10101 | 2 | 0 | 0 | 0 | 1 | 0 | 0 | 0 | 0 | 1 | 0 |
| 13 | 7 | 10101 | 2 | 0 | 0 | 0 | 1 | 0 | 0 | 0 | 0 | 1 | 0 |

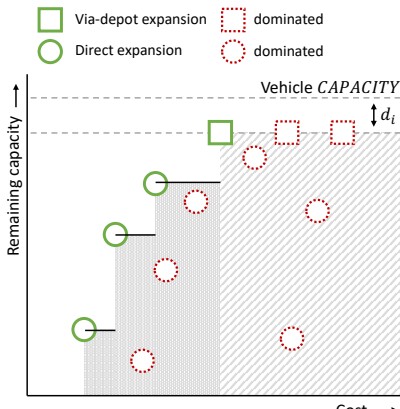

(a) Example beam for VRP with variables, grouped by set of visited nodes (left) and feasible, non-dominated expansions (right), with $2n$ columns corresponding to $n$ direct expansions and $n$ via-depot expansions. Some expansions to unvisited nodes are infeasible, e.g. due to the capacity constraint or a sparse adjacency graph. The shaded areas indicate groups of candidate expansions among which dominances should be checked: for each set of visited nodes there is only one non-dominated via-depot expansion (indicated by solid green square), which must necessarily be an expansion of the solution that has the lowest cost to return to the depot (indicated by the dashed green rectangle ; note that the cost displayed excludes the cost to return to the depot). Direct expansions can be dominated (indicated by red dotted circles) by the single non-dominated via-depot expansion or other direct expansions with the same DP state (set of visited nodes and expanded node, as indicated by the shaded areas). See also Figure 4b for (non-)dominated expansions corresponding to the same DP state.

(b) Example of a set of dominated and non-dominated expansions (direct and via-depot) corresponding to the same DP state (set of visited nodes and expanded node $i$) for VRP. Non-dominated expansions have lower cost or higher remaining capacity compared to all other expansions. The right striped area indicates expansions dominated by the (single) non-dominated via-depot expansion. The left (darker) areas are dominated by individual direct expansions. Dominated expansions in this area have remaining capacity lower than the cumulative maximum remaining capacity when going from left to right (i.e. in sorted order of increasing cost), indicated by the black horizontal lines.

Figure 4: Implementation of DPDP for VRP

### A.2.2 Vehicle Routing Problem

For VRP, the dominance check has two dimensions (cost *and* remaining capacity) and additionally we need to consider $2n$ actions: $n$ direct and $n$ via the depot (see Figure 4a). Therefore, as we will explain, we check dominances in two stages: first we find (for each DP state) the *single* non-dominated 'via-depot' expansion, after which we find all non-dominated 'direct' expansions (see Figure 4b).

The DP state of each expansion is defined by the expanded node (the new current node) and the set of visited nodes. For each DP state, there can be only *one*[12] non-dominated expansion where the last action was via the depot, since all expansions resulting from 'via-depot actions' have the same remaining capacity as visiting the depot resets the capacity (see Figure 4b). To find this expansion, we first find, for each unique set of visited nodes in the current beam, the solution that can return to the depot with lowest total cost (thus including the cost to return to the depot, indicated by a dashed green rectangle in Figure 4a). The single non-dominated 'via-depot expansion' for each DP state must necessarily be an expansion of this solution. Also observe that this via-depot solution cannot be dominated by a solution expanded using a direct action, which will always have a lower remaining vehicle capacity (assuming positive demands) as can bee seen in Figure 4b. We can thus generate the non-dominated via-depot expansion for each DP state efficiently and independently from the direct expansions.

For each DP state, all *direct* expansions with cost higher (or equal) than the via-depot expansion can directly be removed since they are dominated by the via-depot expansion (having higher cost and lower remaining capacity, see Figure 4b). After that, we sort the remaining (if any) direct expansions

---

[12]Unless we have multiple expansions with the same costs, in which case can pick one arbitrarily.

627 for each DP state based on the cost (using a segmented sort as the expansions are already grouped
628 if we generate them similarly to TSP, i.e. per column in Figure 4a). For each DP state, the lowest
629 cost solution is never dominated. The other solutions should be kept only if their remaining capacity
630 is strictly larger than the largest remaining capacity of all lower-cost solutions corresponding to the
631 same DP state, which can be computed using a (segmented) cumulative maximum computation (see
632 Figure 4b).

### A.2.3 TSP with Time Windows

634 For the TSPTW, the dominance check has two dimensions: cost and time. Therefore, it is similar to
635 the check for non-dominated direct expansions for the VRP (see Figure 4b), but replacing remaining
636 capacity (which should be maximized) by current time (to be minimized). In fact, we could reuse
637 the implementation, if we replace remaining capacity by time multiplied by $-1$ (as this should be
638 minimized). This means that we sort all expansions for each DP state based on the cost, keep the first
639 solution and keep other solutions only if the time is strictly lower than the lowest current time for all
640 lower-cost solutions, which can be computed using a cumulative minimum computation.

### A.3 Finding the top $B$ solutions

642 We may generate all 'candidate' non-dominated expansions and then select the top $B$ using the score
643 function. Alternatively, we can generate expansions in batches, and keep a streaming top $B$ using a
644 priority queue. We use the latter implementation, where we can also derive a bound for the score as
645 soon as we have $B$ candidate expansions. Using this bound, we can already remove solutions before
646 checking dominances, to achieve some speedup in the algorithm.[13]

### A.4 Performance improvements

648 There are many possibilities for improving the speed of the algorithm. For example, PyTorch lacks a
649 segmented sort so we use a much slower lexicographic sort instead. Also an efficient GPU priority
650 queue would allow much speedup, as we currently use sorting as PyTorch' top-$k$ function is rather
651 slow for large $k$. In some cases, a binary search for the $k$-th largest value can be faster, but this
652 introduces undesired CUDA synchronisation points. We currently use multiprocessing to solve
653 multiple instances on a single GPU in parallel, introducing a lot of Python overhead. A batched
654 implementation would give a significant speedup.

## B TSP with Time Windows

656 This section contains additional information for the TSPTW.

### B.1 Adaption of model for TSPTW

658 The model updates the edge embedding $e_{ij}^l$ for edge $(i, j)$ in layer $l + 1$ using node embeddings $x_i^l$
659 and $x_j^l$ with the following equation (Equation (5) in [26]):

$$e_{ij}^{l+1} + \text{ReLU}(\text{BatchNorm}(W_3^l e_{ij}^l + W_4^l x_i^l + W_5^l x_j^l)) \tag{2}$$

660 where $W_3^l$, $W_4^l$ and $W_5^l$ are trainable parameters. We found out that the implementation[14] actually
661 shares the parameters $W_4^l$ and $W_5^l$, i.e. $W_4^l = W_5^l$, resulting in $e_{ij}^l = e_{ji}^l$ for all layers $l$, as for
662 $l = 0$ both directions are initialized the same. To allow the model to make different predictions for
663 different directions, we implement $W_5^l$ as a separate parameter, such that the model can have different
664 representations for edges $(i, j)$ and $(j, i)$. We define the training labels accordingly for directed edges,
665 so if edge $(i, j)$ is in the directed solution, it will have a label 1 whereas the edge $(j, i)$ will not (for
666 the undirected TSP and VRP, both labels are 1).

---

[13]This may give slightly different results if the scoring function is inconsistent with the domination rules,
i.e. if a better scoring solution would be dominated by a worse scoring solution but is not since that solution is
removed using the score bound before checking the dominances.

[14]https://github.com/chaitjo/graph-convnet-tsp/blob/master/models/gcn_layers.py

## B.2 Dataset generation

We found that using our DP formulation for TSPTW, the instances by [6] were all solved optimally, even with a very small beam size (around 10). This is because there is very little overlap in the time windows as a result of the way they are generated, and therefore very few actions are feasible as most of the actions would 'skip over other time windows' (advance the time so much that other nodes can no longer be served)[15]. We conducted some quick experiments with a weaker DP formulation, that only checks if actions *directly* violate time windows, but does not check if an action causes other nodes to be no longer reachable in their time windows. Using this formulation, the DP algorithm can run into many dead ends if just a single node gets skipped, and using the GNN policy (compared to a cost based policy as in Section 4.4) made the difference between good solutions and no solution at all being found.

We made two changes to the data generation procedure by [6] to increase the difficulty and make it similar to [10], defining the 'large time window' dataset. First, we sample the time windows around arrival times when visiting nodes in a random order without any waiting time, which is different from [6] who 'propagate' the waiting time (as a result of time windows sampled). Our modification causes a tighter schedule with more overlap in time windows, and is similar to [10]. Secondly, we increase the maximum time window size from 100 to 1000, which makes that the time windows are in the order of 10% of the horizon[16]. This doubles the maximum time window size of 500 used by [10] for instances with 200 nodes, to compensate for half the number of nodes that can possibly overlap the time window.

To generate the training data, for practical reasons we used DP with the heuristic 'cost heat + potential' strategy and a large beam size (1M), which in many cases results in optimal solutions being found.

---

[15]If all time windows are disjoint, there is only one feasible solution. Therefore, the amount of overlap in time windows determines to some extent the 'branching factor' of the problem and the difficulty.

[16]Serving 100 customers in a 100x100 grid, empirically we find the total schedule duration including waiting (the makespan) is around 5000.