# OpenReview forum: "Deep Policy Dynamic Programming for Vehicle Routing Problems"
_NeurIPS.cc/2021/Conference — NeurIPS 2021 Submitted_

### Official Review · Reviewer_3Pkg · 2021-07-13

**Rating:** 6
**Confidence:** 4

**Summary:**

This paper proposes a framework for solving vehicle routing problems by combining the strength of deep learning and dynamic programming. The basic idea is to use a graph neural network to restrict the search space and then boost the DP algorithm. The experiment results for the traveling salesman problem (TSP), the vehicle routing problem (VRP), and TSP with time windows (TSPTW) show that the proposed method can achieve similar performance as LKH and outperform most other neural nets-based algorithms.

**Limitations And Societal Impact:**

Some comments:

1)  I think this framework has very good generalizability and is very promising for solving other problems. I encourage the authors to apply this framework to other graph-related combinatorial optimization problems (for example, job scheduling).

**Main Review:**

Originality: This work employs a framework of using learned neural heuristics to guide dynamic programming algorithms. The model consists of a GNN [3] and a beam search. I think this framework is not completely novel but solid enough.

Quality: The paper is solid. Authors provide experiments with detailed analysis.

Clarity: The paper is clearly written and easy to follow.

Significance: The proposed framework exhibits good generalization towards different vehicle routing problems, as it can efficiently handle hard constraints. Meanwhile, the experiment results demonstrate that the model outperforms the state-of-art algorithm to solve TSPTW, which is impressive. However, it still does not beat some other algorithms [1, 2] for CVRP.

[1] Thibaut Vidal, Teodor Gabriel Crainic, Michel Gendreau, Nadia Lahrichi, and Walter Rei. A hybrid genetic algorithm for multidepot and periodic vehicle routing problems. Operations Research, 60(3):611–624, 2012.

[2] Hao Lu, Xingwen Zhang, and Shuang Yang. A learning-based iterative method for solving vehicle routing problems. In International Conference on Learning Representations, 2020.

[3] Chaitanya K Joshi, Thomas Laurent, and Xavier Bresson. An efficient graph convolutional network technique for the travelling salesman problem. arXiv preprint arXiv:1906.01227, 2019.

**Time Spent Reviewing:**

10

---

> ### Author Response · Authors · 2021-08-16
> **Thank you for your positive review ; some comments on comparisons**
>
> Dear reviewer,
>
> Thank you for appreciating our paper and considering it promising. We will consider solving other problems in future work.
>
> Indeed, we do not beat HGS for CVRP, which is a problem specific SOTA algorithm, but also we do not claim so. We added Lu et al. to our results table for completeness, but please be careful with drawing conclusions: their method trains for each instance and is thus two orders of magnitude slower. Their average cost is computed on a different and smaller test set than ours (and most related works) and therefore cannot be directly compared.

---

### Official Review · Reviewer_C5JT · 2021-07-15

**Rating:** 5
**Confidence:** 4

**Summary:**

The paper proposes a framework for solving vehicle routing problems, based on a combination of supervised learning and dynamic programming. The proposed method is an extension of the restricted dynamic programming algorithm where at each iteration, the selection of the (fixed number of) most promising solutions is done by a learning-based heuristic. The framework is tested on instances of 100 nodes of the TSP, CVRP and TSP with time windows.


**Limitations And Societal Impact:**

Yes, in the checklist.

**Main Review:**

Originality
---------------
The novelty of this work is the use of a learned heuristic to define the criteria upon which the most promising states are selected in a restricted dynamic programming algorithm. The criteria is based on a heat map provided by an existing pretrained neural network and a “potential” formula that is handcrafted by the authors.

The contributions are clearly positioned w.r.t related works.

Quality
----------
The paper is of high quality. The explanation of how the framework is adapted for each of the 3 routing problems is clear.
The authors discuss both the strengths and limitations of their work and do not overclaim their contribution.
I appreciated the comparison to a stronger (non-learned) heuristic baseline (HGS) for the VRP experiments.

Regarding the framework itself, I wonder how it differs from a beam search using the same scoring function (heat+potential) with the elimination of dominated solutions at each step. Or an A* algorithm with the same elimination trick. In other words, what is "dynamic programming-specific" about this framework?

Clarity
---------
The paper is very well-written and easy to follow.

Details:
* I found confusing that at L112 “state variables” are defined as triplets (visited, current, cost) and then  L122 “DP state” is defined as only (visited, current)
* Figure 3 titles and legends fonts are too small
* L189 "travel time is equal to cost/distance" not clear
* L337 We *believe*

Significance
----------------
Th proposed framework is flexible and able to naturally handle constraints, which is a big strength compared to other NCO methods.

However, the reported results are slightly better or close to the best existing NCO methods but still not as good as specialized algorithms. And these results were obtained on graphs of size 100, which is relatively small compared to what can be solved, both with specialized methods and NCO today. For instance the cited reference [17] solves TSP problems with up to 10,000 nodes. As noted by the authors, standard VRP benchmarks have instances of 100 to 1000 nodes.

The authors note that scaling is challenging because of the “fully-connected O(n^2) GNN” but this GNN is an essential ingredient of their framework so it’s not clear if/how this framework can scale.

Therefore despite the high quality of the paper, I feel like it is quite an incremental contribution.






**Time Spent Reviewing:**

6

---

> ### Author Response · Authors · 2021-08-16
> **Thank you for appreciating the paper quality ; some clarifications**
>
> Dear reviewer,
>
> Thank you for your review and for appreciating the high quality of the paper.
>
> By removing dominated solutions, you keep only the best solution for each (DP) state, which *is* by definition dynamic programming. Therefore, what you describe is effectively DPDP, although DPDP performs dominance checks 'locally' and thus much more efficiently then a 'naïve implementation', by grouping states by sets of visited nodes before expanding (see Appendix A2). While this may seem incremental from the beam search perspective, the connection to DP algorithms is in fact part of our contribution, and the improvements over both existing restricted dynamic programming algorithms and plain beam search are significant.
>
> Thanks for your comments regarding clarity. For each state, which is uniquely defined by (visited, current), we keep track of the variables (visited, current, cost), which we agree is a bit confusing and we will try to clarify.
>
> To scale the method and release the O(n^2) model scaling issue, we think a promising direction would be to look at sparse GNN architectures.

---

> > ### Comment · Reviewer_C5JT · 2021-08-27
> > **Thanks for the response**
> >
> > I thank the authors for the clarifications. I do not have further questions.

---

### Official Review · Reviewer_jZhT · 2021-07-15

**Rating:** 5
**Confidence:** 4

**Summary:**

The paper investigates dynamic programming techniques for routing that relies on auxiliary neural network models. The idea is to predict a "heatmap" of edges indicating whether they are promising or not for good-quality solutions, which needs to be generated only once per instance. The problem is then solved by a (forward) dynamic program that expands states based on scores extracted by the heatmap. The methodology is applied to the vehicle routing problem (VRP), the traveling salesperson problem (TSP), and the TSP with time windows (TSPTW).

**Limitations And Societal Impact:**

No direct negative societal impacts to the best of my knowledge.



**Main Review:**

**Originality**

I believe the key ideas underlying the paper are somewhat incremental, albeit nicely combined together. More specifically, the methodology is conceptually close to a value-function approximation that is defined in terms of their scoring policy. Thus, it is akin to a reinforcement learning approach and builds upon many of the references that the authors appropriately cite but do not discuss or highlight appropriately, in my view.

In particular, deep learning has been used to guide tree search (e.g., references [26] and [34] in the paper). The differences here, to the best of my knowledge, are (a) the routing applications, (b) the fact that authors expand DP states (allowing them to break some symmetry and dominance) and not tree nodes; and (c) the heatmap and beam search implementation for DPs. I would strongly suggest the authors to extend their related work discussion to make this comparison clearer.

**Quality**

My major concern in terms of quality is that results still read as somewhat preliminary (which seems to be still consistent with the previous ICML comments as well).

Indeed, there are improvements with respect to the neural network approaches evaluated by the authors and the TSPTW results are meaningful. However, it does not seem that the paper outperforms exact approaches for small instances (e.g., Concorde/TSP) nor standard heuristic techniques (e.g., LKH/VRP) in several settings. The overall tradeoffs between their technique (considering the implementation complexities/training) and other methods are not clear.

Moreover, the scalability benefits mentioned by the authors was not investigated further; for example, authors do not indicate whether their method really scales for instances with significant more nodes (e.g., a few thousand).

**Clarity**

The paper is well written. I believe the results are mostly reproduceable, though there are a few parameters based on "educated guesses" as indicated by the authors. Some further discussion concerning training and limitations could have also been beneficial.

**Significance**

In my view, the paper presents meaningful and interesting ideas, but still requires a stronger experimental section to be of interest beyond a specific area (i.e., deep learning models to routing problems). For example, authors could demonstrate the scalability of their method to instances with a few thousand nodes, which is common in papers investigating LKH-3. Connection to existing works must also be presented in much more details.

**Time Spent Reviewing:**

3

---

> ### Author Response · Authors · 2021-08-16
> **Thank you for your helpful suggestions ; some comments**
>
> Dear reviewer,
>
> Thank you for your review which contains helpful suggestions.
>
> * We will improve the comparison to tree search based methods, which you correctly summarised.
> * We aim to present a general approach and evaluate it on different problems, where we find reasonable performance across problems, but our goal (and claim) is not to outperform problem-specific SOTA solvers for TSP/VRP. However, as these problems are widely studied, they are useful to give a good impression of the performance DPDP as a more general strategy can achieve.
> * Scaling to larger problems will likely involve more problem-specific engineering (such as sparsification), which we consider out of the scope of the current paper. We think however DPDP is also interesting for smaller problems with more constraints (such as TSPTW), which are more difficult for search based approaches.
>
> We will try to improve unclarities regarding implementation/training, e.g. total data generation/training times etc. (see also [26]). For DPDP using the trained model, we describe the implementation in general, as well as in detail in the Appendix and we will provide code. The tradeoffs in terms of (test time) running time vs performance should be clear from the graphs. Could you comment if you find anything else unclear?

---

> > ### Comment · Reviewer_jZhT · 2021-08-23
> > **Feedback**
> >
> > Thank you for the feedback, I appreciate the reply.

---

### Official Review · Reviewer_fh68 · 2021-07-16

**Rating:** 4
**Confidence:** 3

**Summary:**

The present submission is concerned with dynamic programming (DP) methods for  traveling salesperson problems and a capacitated vehicle problem. Since DP for these problems struggles with enormous DP table sizes, the authors propose to heuristically prune the DP table using a neural network.

Essentially, the idea is to leverage recent approaches that use neural network to predict good solutions for combinatorial optimization problems. In this particular case, a neural network is used to predict partial solutions, and these in turn allow the others to compute priority values for the edges. High priority edges are preferred in the solution expansion step of the DP.

The contribution of the paper is the evaluation of the approach (which, according to the authors, has not been tried before); the neural network itself stems from the literature (with few modifications).

**Limitations And Societal Impact:**

My conclusion from the experiments is that the proposed approach is promising, but I am not convinced that it would beat mathematical programming based approaches, even though these would certainly be the method of choice in combinatorial optimization/operations research where TSP/VRP originates and is heavily studied. Indeed, the only math. programming approach (Concorde) included in the comparison turns out to be superior. To illuminate this question, further literature review and extended benchmarks with new instance sets from the OR community would be required.

**Main Review:**

The paper is generally well written and nicely illustrated. The only two aspects of the writeup that I found lacking was that none of the considered problems are defined in the paper, not even informally; and that not all column labels in the computational tables are defined.

I found the proposed approach intriguing. From the combinatorial optimization point of view, I feel it is more attractive to combine DP and neural networks than to use neural networks directly to predict good solutions. Additionally, it seems that the approach is easily applied to other problems as well, provided there is a DP scheme and a suitable neural network; nothing in the approach is really specific to vehicle routing/TSP. On the other hand, the approach does away with a major benefit of DP, namely, that optimality may be guaranteed. In fact, there are no theoretical guarantees.

The computational experiments are extensive:

- For the TSP, the authors compare to a large body of literature including the Concorde TSP solver which is a staple in the combinatorial optimization community. Indeed, the results look favorable for Concorde (Concorde finds a better solution while requiring only one fourth of the running time), although the authors approach seems to be superior to the other algorithms from the literature.

- For the VRP, the authors compare to genetic algorithms and an ancient DP approach. The authors approach offers a speed-up of about a factor of 10 at the expense of losing 2 percent of the solution quality. The approach is unable to find competitive solutions even if allowed several times the running time of the approaches from the literature. Hence, the usefulness of the approach depends on the trade-offs that the user is willing to make.

- TSP with time windows: The authors compare against a combination of constraint programming and learning, and against a local search appraoch. This time, the results look largely favorable for the new approach.

Overall, however, the choice of reference algorithms seems a bit arbitrary; I found the text did not make clear why were these particular algorithms chosen and others were not. For instance, reference [1] is not included in the VRP experiments. The benchmark instances for TSP stem from another learning publication; what about the standard TSPlib benchmark?



**Time Spent Reviewing:**

3

---

> ### Author Response · Authors · 2021-08-16
> **Thank you for your feedback ; please consider our work in scientific context**
>
> Dear reviewer,
>
> Thank you for your appreciation of our method and useful feedback which we will use to improve the paper.
>
> We would like to emphasize that it is not our goal to outperform problem specific mathematical programming approaches, which are currently still preferred practically. However, the performance gap with learning based methods is rapidly decreasing and our work contributes to this by presenting a new and general idea to combine learning and mathematical programming.
>
> Please consider our work in relation to the line of work on learning for combinatorial optimization. We think we make an useful contribution in this promising research direction.
>
> Answers and minor comments:
> * We will add missing problem descriptions and clarify table columns (which are the same as in related work, e.g. [29])
> * Indeed we lose the optimality (as does any practical method) but in a clever way. Note that there still is an asymptotic guarantee for a finite beam size.
> * We did not realize an implementation of [1] was available, which is very recent. We will try to add the comparison, but note that we do not claim to outperform SOTA methods specific to CVRP, so it is not needed to support our claims. Nevertheless, we already included comparison to the very recent HGS implementation as a reference.
> We follow previous works on learning methods for evaluation but we agree it would be good to add TSPlib. We do not think it would significantly affect our conclusions.

---

### Author Response · Authors · 2021-08-16
**Thank you for your reviews ; clarification of goal/claims which is not to directly outperform SOTA TSP/VRP but provide new general method**

Dear reviewers,

Thank you for your time reviewing our paper. We are sorry for our delayed response. If possible, please have a look at the individual responses.

Summarizing, it seems all reviewers appreciate the quality and clarity of the paper.

The major concern seems to be the importance of the results in relation to SOTA problem-specific solvers for TSP/VRP. We want to emphasize that our goal (and claim) is not to outperform such problem-specific solvers, which are the result of decades of research. Matching this performance is still a challenge for all papers in this research direction, although the gap has been rapidly decreasing.

We aim to contribute by providing a new method that is relatively simple and general to be applied to different problems and we believe there is room to address current limitations such as scalability in future work. Therefore, we think our work presents a useful contribution in this line of research.

---

### Decision · Program_Chairs · 2021-09-27

**Decision:**

Reject

**Comment:**

The paper presents a neural network based approach to dynamic programming with large state spaces and applies it to traveling salesman and vehicle routing problems. While the paper is well written and interesting, the experimental side is preventing acceptance at NeurIPS. In particular, while it is not strictly needed to outperform competitors, the theoretical novelty just needs to be larger to merit publication.

In order to better prove the merit of the presented approach, running the algorithm on problems where dynamic programming is a better methodology than TSP might be one avenue, as suggested by the reviewers.